# Effects of plyometric training on jump, sprint, and change of direction performance in adolescent soccer player: A systematic review with meta-analysis

**Tianjing Zheng[1‡], Runzhou Kong[2‡], Xiaowen Liang[3‡], Zhilong Huang[4], Xicai Luo[1], Xuan Zhang[5], Yichao Xiao [5]\***

**1** Guangzhou Railway Polytechnic, Guangzhou, China, **2** Guangzhou Sport University, Guangzhou, China, **3** Guangzhou Children's Palace, Guangzhou, China, **4** Zhongshan Torch Polytechnic, Zhongshan, China, **5** Liuzhou Institute of Technology, Liuzhou, China

‡ On behalf of the co-first authors.
* xiaoyc9@163.com

## Abstract

### Background

Soccer requires a high level of physical fitness, particularly in jumping, sprinting, and change-of-direction (COD) performance. Plyometric training has been extensively studied in adult athletes, but its effects on these abilities in adolescent soccer players remain insufficiently evaluated.

### Objective

This systematic review with meta-analysis examined the effects of plyometric training on jump, sprint and COD performance in adolescent soccer player.

### Methods

Eligible randomized controlled trials were identified through searches of PubMed, Web of Science, Scopus, and SPORTDiscus databases, focusing solely on published studies. Study quality was assessed using the PEDro scale, and statistical analysis was conducted using Stata software to calculate the standardized mean difference (SMD) and 95% confidence intervals.

### Results

Twenty studies comprising 28 randomized controlled trials with a total sample size of 796 participants were included. The results indicated that plyometric training significantly improved the jumping ability (SMD = 0.76, 95%CI: [0.59, 0.93]; moderate effect), sprinting ability (SMD = -0.45, 95%CI: [-0.57, -0.32]; small effect), and COD (SMD = -0.76, 95%CI: [-1.04, -0.47]; moderate effect) of adolescent soccer players.

**Data availability statement:** All relevant data are within the article and its Supporting Information files.

**Funding:** Guangdong Province Science and Technology Innovation Strategy Special Fund (Grant Number: pdjh2023b0949).

**Competing interests:** The authors have declared that no competing interests exist.

## Conclusion

Plyometric training effectively enhances jumping, sprinting, and COD abilities in adolescent soccer players. Compared to soccer-specific training alone, PT demonstrated moderate improvements in jumping and COD performance and small improvements in sprinting ability. These findings highlight the importance of incorporating PT into routine soccer training regimens to develop explosive strength and agility in adolescent athletes.

## Introduction

Soccer is one of the most popular sports worldwide, particularly among youth, with participation rates growing steadily [1]. In high-level competitions, high-intensity actions such as jumping, sprinting, and changes of direction are not only critical abilities for creating scoring opportunities [2], but also key indicators distinguishing players of different competitive levels [3]. The execution of these high-intensity movements heavily relies on rapid force generation and high power output by the muscles, with the stretch-shortening cycle (SSC) being regarded as the core physiological mechanism underpinning these abilities [4]. Given that adolescence is a critical window for rapid neuromuscular development, designing scientifically rigorous and practical training interventions for this stage is paramount importance [5,6].

Among various training methods, plyometric training is considered a highly effective strategy for utilizing and optimizing the stretch-shortening cycle (SSC) mechanism. It has been widely implemented in the physical conditioning programs of youth soccer players [7,8]. Previous research has demonstrated that plyometric training significantly improves the maximum strength [9], jumping ability [7,10], acceleration [8,11], speed [8,12], and change-of-direction (COD) capacity [13,14] of youth soccer players while also contributing to a reduction in the risk of sports injuries [15]. The core mechanism of plyometric training lies in effectively utilizing the SSC mechanism. During this process, the muscle is rapidly elongated during the eccentric phase, during which the elastic tissues of the muscle-tendon complex store mechanical energy. Simultaneously, the stretch reflex is activated, enhancing the nervous system's efficiency in recruiting motor units [16–18]. When the muscle transitions swiftly from a stretched to a shortened state, the stored mechanical energy is released and works in concert with active muscle contraction to significantly enhance force output and instantaneous power generation [19,20]. This process optimizes the mechanical properties of the muscle-tendon complex and improves neural regulation efficiency, thereby providing a robust physiological foundation for enhanced athletic performance.

Evidence suggests that concentric peak force (CPPF) and isometric maximal force (IMF) in youth soccer players increase linearly with age, reflecting a synergy between strength development and neuromuscular maturation, offering more significant potential for training adaptations to plyometric training interventions [21]. While soccer-specific training may improve the physical fitness of young athletes,

plyometric training provides additional neuromuscular stimulation, further promoting their long-term athletic development [5,22]. Moreover, plyometric training induces neuromuscular adaptations (e.g., enhanced motor unit recruitment efficiency and improved neural conduction velocity), which align closely with the naturally occurring neuromuscular adaptations during adolescence [6]. This synergy further enhances the responsiveness of youth athletes to plyometric training.

In recent years, there has been growing research interest in the specific effects of plyometric training on youth soccer players, with preliminary evidence supporting its efficacy. However, these studies exhibit limitations in sample selection and research design. For example, Ramirez-Campillo et al.'s study [2] included adult soccer players up to 23 years of age, lacking a strict focus on youth players. Chen et al.'s study [6] only evaluated CMJ (countermovement jump) and 20-meter sprint, overlooking multi-dimensional aspects of speed development (e.g., acceleration and top speed) and omitting evaluations of various types of jumps and COD abilities. Additionally, existing research still lacks a comprehensive assessment of different types of jumps, sprints, and COD abilities in youth soccer players, limiting our understanding of plyometric training's potential for neuromuscular adaptation. Therefore, this study aims to systematically assess the overall effects of plyometric training on jumping, sprinting, and COD abilities in youth soccer players.

## Methods

### Literature search

This review is based on the Preferred Reporting Items for Systematic Reviews and Meta-Analyses (PRISMA) checklist [23]. The inclusion criteria were defined using the PICOS framework (Population, Intervention, Comparators, Outcomes, and Study Design) [24]. The study has been registered in the International Prospective Register of Systematic Reviews (PROSPERO: CRD42024579445). To comprehensively identify relevant studies, a search was conducted across four databases—PubMed, Web of Science (all databases), Scopus, and SPORTDiscus—from the inception of each database up to August 7, 2024. Boolean operators (OR, AND) were utilized with a series of keywords, which were finalized based on literature reviews, meta-analyses, expert opinions, and the MeSH Database. Two reviewers (TJ and RZ) conducted the literature search and screening process independently. In cases where discrepancies arose, a third reviewer (XW) provided a final decision to ensure consistency. Furthermore, no language restrictions were applied, and grey literature was excluded from the search strategy. The specific search string was as follows: ("Lower limb explosive strength" OR "Explosive strength" OR "explosive force" OR "Explosive power" OR "power" OR "Countermovement jump" OR "CMJ" OR "squat jump" OR "SJ" OR "standing long jump" OR "SLJ" OR "drop jump" OR "DJ" OR "sprint performance" OR "10m" OR "20m" OR "30m" OR "50m" OR "vertical jump" OR "VJ" OR "change of direction" OR "COD") AND ("plyometric" OR "plyometrics" OR "PT" OR "pliometrique" OR "entrainement pliometrique" OR "salto pliome´trico" OR "velocidad") AND ("Adolescent" OR "Adolescents" OR "Adolescence" OR "Teens" OR "Teen" OR "Teenagers" OR "Teenager" OR "Youth" OR "Youths" OR "Female Adolescent" OR "Female Adolescents" OR "Male Adolescent" OR "Male Adolescents" OR "Child" OR "Children") AND ("soccer"). The screening process is depicted in Fig 1.

### Eligibility criteria

The detailed inclusion and exclusion criteria are shown in Table 1.

### Data extraction

Data from the included studies and participant characteristics were extracted by one author (TJ) and compiled in Microsoft Excel. After the data collection, another author (ZL) verified and confirmed the extracted content. In cases where the extracted information was contentious, a third co-author (XW) made the final determination. This study extracted information on the authors of the included studies (first author's name and publication year), characteristics of the study population (gender, age, sample size), and intervention variables (intervention duration, intervention period, intervention

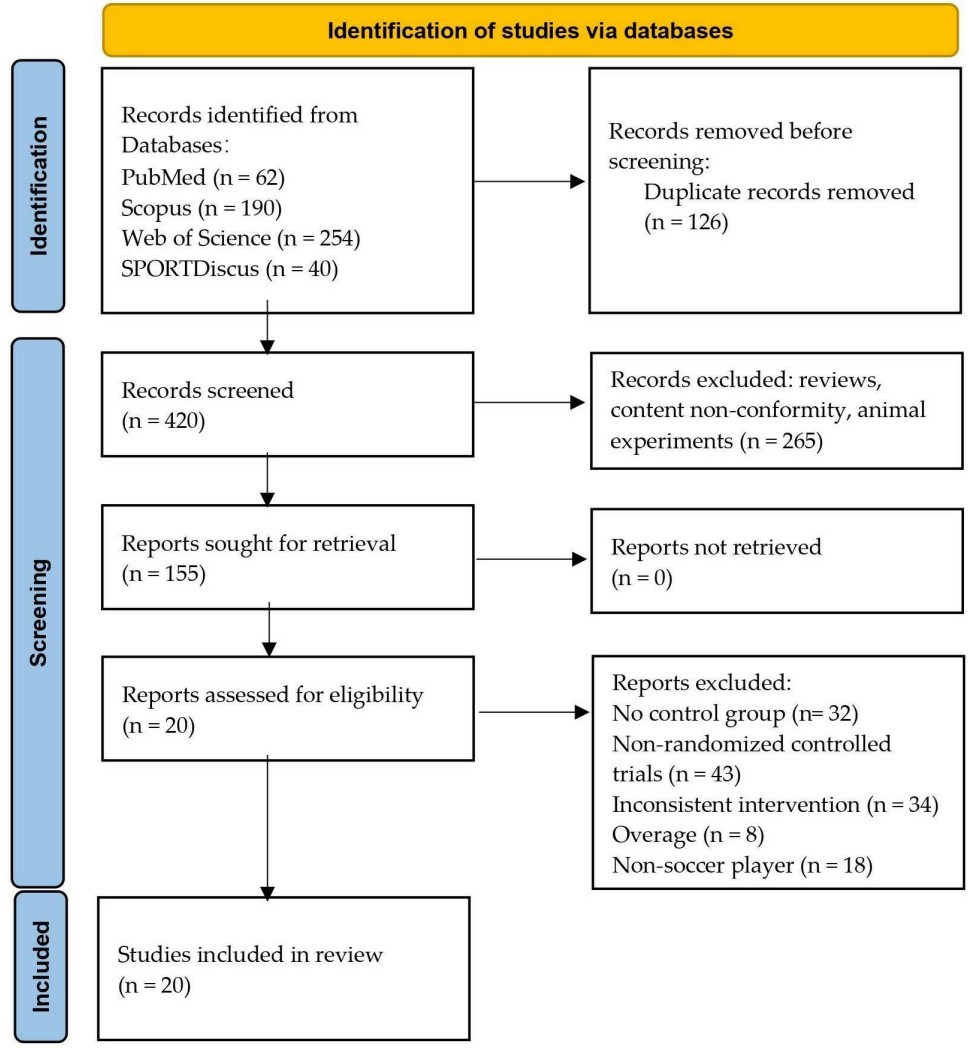

**Fig 1. Visual Overview of the Literature Screening and Inclusion Pathway.**

frequency). When data were presented as bar charts or error bars, Getadata software was used for data extraction. Positive standardized mean difference (SMD) values in interpreting effect sizes reflect improvements in abilities such as jumping. For measures where lower times indicate better performance, such as sprinting and change-of-direction, negative SMD values signify reduced completion times, corresponding to enhanced performance. If data could not be extracted, attempts were made to contact the corresponding authors or reach out via ResearchGate. Should the data remain inaccessible before the article's publication, the literature in question would be excluded after three attempts to contact the corresponding authors within two weeks. The extracted metrics are detailed in Table 2.

**Methodological quality and risk of bias**

The methodological quality of all included studies in this research was assessed using the PEDro scale (Physiotherapy Evidence Database), with scores ranging from 0 to 10, where a higher score indicates a higher quality of the included literature. The validity and reliability of the PEDro scale have been established previously [28]. The quality rating scale for

**Table 1. Eligibility Criteria for Inclusion and Exclusion of Studies.**

| Category | Inclusion criteria | Exclusion Criteria |
|---|---|---|
| Population | The study population consists exclusively of adolescent soccer players, both male and female, with the age range defined as 10 to 18.99 years old | Participants who are not healthy or do not fall within the specified age range for adolescent soccer players. |
| Intervention | Studies included should involve participants engaging in plyometric training either solely or in conjunction with their regular soccer training. | Studies where participants undertake additional training regimens, such as resistance training or high-intensity interval training, alongside plyometric training. |
| Comparator | The control group should only engage in regular soccer training. | Studies lacking a control group. |
| Outcome | Outcome measures were selected to reflect jumping, sprinting, and change-of-direction capabilities, based on their proven reliability and strong relevance to the physical demands of soccer. Studies were deemed eligible for inclusion if they assessed at least one of these indicators [25–27]. | Studies without baseline data or where the full text is inaccessible. |
| Study design | Randomized controlled trials | Cross-sectional studies, case studies. |

**Table 2. Categories and Metrics for Sport Performance Assessment.**

| Outcome categories | Measure |
|---|---|
| Jumping Performance | Countermovement Jump, CMJ |
|  | Squat Jump, SJ |
|  | Standing Long Jump, SLJ |
| Sprinting Performance | 10M-Sprint, 10M |
|  | 20M-Sprint, 20M |
|  | 30M-Sprint, 30M |
| Change of Direction Performance | Illinois |
|  | T-Test |
|  | Zig Zag Drill |

the PEDro scale is categorized as poor (<4), fair (4–5), good (6–8), and excellent (9–10). The quality of all documents was independently evaluated by two co-authors, with any discrepancies or uncertainties resolved by a third co-author. Publication bias risk was visually inspected using funnel plots and quantified using Egger's test. In the event of publication bias, the missing literature was supplemented using the trim-and-fill method, which identifies asymmetry in funnel plots and estimates missing studies to adjust for potential publication bias.

## Statistical analysis

All data in this study were statistically analyzed using Stata software. The three capabilities assessed in this study—jumping ability, sprinting ability, and change of direction ability—each consist of three different metrics. Therefore, the Standardized Mean Difference (SMD) was selected as the summary effect measure, with the SMD value and its 95% confidence interval presented together. The SMD values can be interpreted as follows: trivial (SMD < 0.20), small (0.20 ≤ SMD < 0.50), moderate (0.50 ≤ SMD < 0.80), and large (SMD ≥ 0.80) [29]. Initially, the data included in this study were the baseline and post-test mean values and standard deviations for the experimental and control groups. However, to better evaluate the effects between the two groups after the experimental intervention, we artificially transformed the data into change scores and standard deviations, with the calculation formula $SD_{E,change} = \sqrt{SD_{E,baseline}^2 + SD_{E,final}^2 - (2 \times Corr \times SD_{E,baseline} \times SD_{E,final})}$ [30]. The I² statistic was used to assess the

heterogeneity of the studies; when the $I^2$ value is less than 25%, the heterogeneity can be considered negligible, and a fixed-effect model was used for data analysis. When the $I^2$ value is between 25% and 75%, the studies are considered to have moderate heterogeneity, and a random-effects model was used for data analysis. When the $I^2$ value exceeds 75%, indicating high heterogeneity, a random-effects model was also employed for data analysis [31]. A P-value of less than 0.05 was considered to indicate statistical significance.

## Results

### Description of the included studies

A total of 20 studies were included in this research, encompassing 28 randomized controlled trials, with each trial representing a different intervention condition or control group within the studies. The overall sample size was 796 participants, all of whom were adolescent soccer players, with the age range of 10 to 18.99 years used as an inclusion criterion across the studies. The intervention for the experimental group consisted of plyometric training, with variations in protocols across studies in terms of exercise type, intensity, duration, and frequency. In some studies, plyometric training was integrated into regular soccer sessions; in others, it was conducted separately. The control group engaged solely in soccer training without any additional training interventions. The duration of all studies ranged from 6 to 12 weeks, with intervention frequencies varying from one to three times per week (Table 3).

### Methodological quality and risk of bias assessment

The average PEDro scale score of all included studies was 7.55, with the lowest study scoring 7, indicating "good" methodological quality, and four studies scoring 9, indicating "excellent" quality (Table 4). Additionally, to provide a more comprehensive assessment of bias risk, we employed the Cochrane Risk of Bias 2 (ROB2) tool to evaluate each domain of bias in the included studies systematically. The results of the ROB2 assessment are shown in Fig 2 and Table 5. Overall, 60% of the studies were rated as low risk of bias, 35% as having some concerns, and 5% as high risk. These additional analyses provide a complete overview of potential biases across the included studies.

The funnel plot is depicted in Fig 3. Visual inspection of the funnel plot could not quantify publication bias, Hence Egger's test, used to detect asymmetry in funnel plots and assess the presence of publication bias, was employed for a quantitative analysis. The Egger's test revealed the presence of publication bias for jumping ability (t = 2.93, p = 0.005) and change of direction ability (t = -2.42, p = 0.032), while no publication bias was detected for sprinting ability (p = 0.059). The trim-and-fill method was applied to address the identified publication bias. After the adjustment, two additional studies were imputed for jumping ability, and the final results remained consistent with the original findings (p = 0.000). One additional study was imputed for change of direction ability, and the final results were consistent with the original findings (p = 0.000). These findings suggest that, despite the presence of publication bias, the original outcomes remain robust.

### Meta-analysis results

This study incorporated data from 47 randomized controlled trials involving 1,263 participants and compared the effects of plyometric training with regular soccer training on the jumping ability of adolescent soccer players. The analysis included three jumping ability indicators: CMJ, SJ, and SLJ (Fig 4). The results demonstrated that plyometric training significantly outperformed regular soccer training in improving jumping ability, with a moderate positive overall effect size (SMD = 0.76, 95% CI: [0.59, 0.93], p < 0.001). Planned subgroup analyses further revealed that plyometric training had a significant positive impact on all three jumping ability indicators compared with regular soccer training: CMJ (SMD = 0.80, 95%CI: [0.55, 1.06], p < 0.001), SJ (SMD = 0.63, 95%CI: [0.30, 0.95], p < 0.001), and SLJ (SMD = 0.84, 95%CI: [0.54, 1.14], p < 0.001).

This study incorporated data from 34 randomized controlled trials involving 990 participants and compared the effects of plyometric training with regular soccer training on the sprinting ability of adolescent soccer players. The analysis

**Table 3. Summary of Study Demographics, Training Parameters, and Outcome Measures.**

| Studies, years | Gender | Age (Years) | | Sample | | Training Duration (Minutes) | Frequency (Times per week) | Period (Weeks) | Measures |
|---|---|---|---|---|---|---|---|---|---|
| | | PT Group | Control Group | PT Group | Control Group | | | | |
| Vera-Assaoka et al [32] | Male | 11.2±0.8 | 11.5±0.9 | 16 | 16 | 21 | 2 | 7 | CMJ, 20M, Illinois |
| Vera-Assaoka et al [32] | Male | 14.4±1.0 | 14.5±1.1 | 22 | 22 | 21 | 2 | 7 | CMJ, 20M, Illinois |
| Sedano et al [11] | / | 18.4±1.1 | 18.2±0.9 | 11 | 11 | / | 2 | 10 | CMJ, SJ, 10M |
| Sammoud et al [12] | Male | 12.7±0.3 | 12.8±0.3 | 11 | 11 | 35-40 | 3 | 12 | 10M, 20M, 30M |
| Ramirez-Campillo et al [33] | Male | 16.9±0.7 | 17.1±0.5 | 12 | 12 | 20 | 2 | 7 | CMJ, SJ, SLJ, 20M, Illinois |
| Ramirez-Campillo et al [33] | Male | 17.1±0.3 | 17.1±0.5 | 14 | 12 | 20 | 2 | 7 | CMJ, SJ, SLJ, 20M, Illinois |
| Ramirez-Campillo et al [34] | Male | 13.9±1.9 | 13.7±1.6 | 25 | 24 | 13 | 2 | 7 | CMJ, 20M, Illinois |
| Ramirez-Campillo et al [34] | Male | 13.1±1.7 | 13.7±1.6 | 24 | 24 | 13 | 2 | 7 | CMJ, 20M, Illinois |
| Ramirez-Campillo et al [35] | Male | 13.2±1.8 | 13.5±1.9 | 19 | 20 | 20 | 2 | 7 | CMJ, 20M, Illinois |
| Ramirez-Campillo et al [36] | Male | 12.9±1.9 | 12.6±1.8 | 8 | 7 | 10-15 | 2 | 8 | CMJ, SLJ, 30M, ZIG ZAG DRILL |
| Padrón-Cabo et al [37] | Male | 12.60±0.70 | 12.39±0.56 | 10 | 10 | 20-35 | 2 | 6 | CMJ, SJ, 10M, 20M |
| Nurper et al [38] | Female | 18.3±2.6 | 18.0±2.0 | 9 | 9 | 60 | 1 | 8 | CMJ, SLJ, 20M |
| Negra et al [39] | Male | 12.7±0.3 | 12.8±0.3 | 11 | 11 | 35-40 | 2 | 4 | CMJ, SJ, SLJ, 20M, Illinois |
| Negra et al [39] | Male | 12.7±0.12 | 12.8±0.12 | 11 | 11 | 35-40 | 2 | 8 | CMJ, SJ, SLJ, 20M, Illinois |
| Negra et al [39] | Male | 12.7±0.17 | 12.8±0.17 | 11 | 11 | 35-40 | 2 | 12 | CMJ, SJ, SLJ, 20M, Illinois |
| Negra et al [14] | Male | 12.7±0.2 | 12.7±0.2 | 13 | 11 | 25-35 | 2 | 8 | 20M, T-TEST |
| Michailidis et al [40] | Male | 10.6±0.6 | 10.6±0.5 | 24 | 21 | 20-25 | 2 | 12 | CMJ, SJ, 10M, 20M, 30M |
| Jlid et al [41] | Male | 11.8±0.4 | 11.6±0.5 | 14 | 14 | 20-25 | 2 | 8 | CMJ, SJ, T-TEST |
| Hammami et al [42] | Male | 15.7±0.2 | 15.8±0.2 | 15 | 13 | 35 | 2 | 8 | 10M, 20M, 30M |
| Hammami et al [9] | Male | 15.7±0.2 | 15.8±0.2 | 14 | 12 | 35 | 2 | 8 | CMJ, SJ |
| Drouzas et al [43] | Male | 10.0±0.5 | 10.2±1.6 | 23 | 22 | 15 | 2 | 10 | CMJ, 10M, 20M |
| Chtara et al [44] | Male | 13.6±0.3 | 13.6±0.3 | 10 | 10 | 20 | 2 | 6 | 10M, 30M, ZIG ZAG DRILL |
| Asadi et al [7] | Male | 11.5±0.8 | 11.7±0.4 | 10 | 10 | 30-40 | 2 | 6 | CMJ, SLJ, 20M |
| Asadi et al [7] | Male | 14.0±0.7 | 14.2±0.6 | 10 | 10 | 30-40 | 2 | 6 | CMJ, SLJ, 20M |
| Asadi et al [7] | Male | 16.6±0.6 | 16.2±0.3 | 10 | 10 | 30-40 | 2 | 6 | CMJ, SLJ, 20M |
| Sammoud et al [12] | Male | 12.7±0.2 | 11.8±0.4 | 13 | 14 | / | 2 | 8 | CMJ, SLJ |
| Liu et al [45] | Male | 16.3±0.6 | 16.3±0.6 | 17 | 17 | 11 | 2 | 8 | CMJ, SLJ, 10M |
| Liu et al [45] | Male | 16.3±0.6 | 16.3±0.6 | 17 | 17 | 14 | 2 | 8 | CMJ, SLJ, 10M |

**Table 4. Quality Assessment of All Included Studies Based on the PEDro Scale.**

| Studies | PEDro Scale Items* | | | | | | | | | | | PEDro Score |
|---|---|---|---|---|---|---|---|---|---|---|---|---|
| | 1 | 2 | 3 | 4 | 5 | 6 | 7 | 8 | 9 | 10 | 11 | |
| Vera-Assaoka et al [32] | 1 | 1 | 1 | 1 | 0 | 0 | 0 | 1 | 1 | 1 | 1 | 8 |
| Sedano et al [11] | 1 | 1 | 0 | 1 | 0 | 0 | 0 | 1 | 1 | 1 | 1 | 7 |
| Sammoud et al [12] | 1 | 1 | 0 | 1 | 0 | 0 | 0 | 1 | 1 | 1 | 1 | 7 |
| Ramirez-Campillo et al [33] | 1 | 1 | 1 | 1 | 1 | 0 | 0 | 1 | 1 | 1 | 1 | 9 |
| Ramirez-Campillo et al [34] | 1 | 1 | 1 | 1 | 1 | 0 | 0 | 1 | 1 | 1 | 1 | 9 |
| Ramirez-Campillo et al [35] | 1 | 1 | 0 | 1 | 0 | 0 | 0 | 1 | 1 | 1 | 1 | 7 |
| Ramirez-Campillo et al [36] | 1 | 1 | 1 | 1 | 1 | 0 | 0 | 1 | 1 | 1 | 1 | 9 |
| Padrón-Cabo et al [37] | 1 | 1 | 1 | 1 | 0 | 0 | 0 | 1 | 1 | 1 | 1 | 8 |
| Nurper et al [38] | 1 | 1 | 0 | 1 | 0 | 0 | 0 | 1 | 1 | 1 | 1 | 7 |
| Negra et al [39] | 1 | 1 | 0 | 1 | 0 | 0 | 0 | 1 | 1 | 1 | 1 | 7 |
| Negra et al [14] | 1 | 1 | 0 | 1 | 0 | 0 | 0 | 1 | 1 | 1 | 1 | 7 |
| Michailidis et al [40] | 1 | 1 | 0 | 1 | 0 | 0 | 0 | 1 | 1 | 1 | 1 | 7 |
| Jlid et al [41] | 1 | 1 | 0 | 1 | 0 | 0 | 0 | 1 | 1 | 1 | 1 | 7 |
| Hammami et al [42] | 1 | 1 | 0 | 1 | 0 | 0 | 0 | 1 | 1 | 1 | 1 | 7 |
| Hammami et al [9] | 1 | 1 | 0 | 1 | 0 | 0 | 0 | 1 | 1 | 1 | 1 | 7 |
| Drouzas et al [43] | 1 | 1 | 0 | 1 | 0 | 0 | 0 | 1 | 1 | 1 | 1 | 7 |
| Chtara et al [44] | 1 | 1 | 0 | 1 | 0 | 0 | 0 | 1 | 1 | 1 | 1 | 7 |
| Asadi et al [7] | 1 | 1 | 0 | 1 | 0 | 0 | 0 | 1 | 1 | 1 | 1 | 7 |
| Sammoud et al [12] | 1 | 1 | 1 | 1 | 0 | 1 | 0 | 1 | 1 | 1 | 1 | 9 |
| Liu et al [45] | 1 | 1 | 1 | 1 | 0 | 0 | 0 | 1 | 1 | 1 | 1 | 8 |

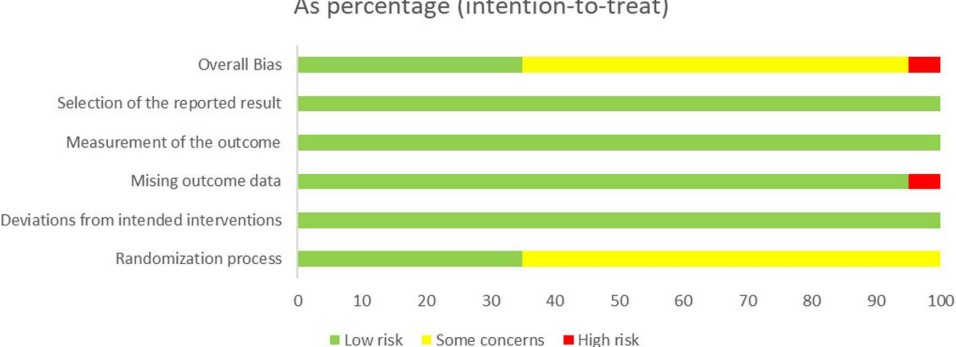

**Fig 2. Risk of Bias of Randomized Clinical Trials.**

included three sprinting ability indicators: 10-meter sprint, 20-meter sprint, and 30-meter sprint (Fig 5). The results demonstrated that plyometric training significantly improved sprinting ability compared with regular soccer training, with a small but positive overall effect size (SMD = -0.45, 95%CI: [-0.57, -0.32], p < 0.001). Planned subgroup analysis revealed that plyometric training positively influenced all three sprinting indicators compared with regular soccer training: 10-meter sprint (SMD = -0.57, 95%CI: [-0.82, -0.33], p < 0.001), 20-meter sprint (SMD = -0.37, 95%CI: [-0.54, -0.21], p < 0.001), and 30-meter sprint (SMD = -0.50, 95%CI: [-0.85, -0.15], p = 0.005).

**Table 5. Study Level Risk of Bias Assessment Using Cochrane Risk ofBias tool 2.0 for Assessing Risk of Bias of Randomized Clinical Trials.**

| | Randomization process | Deviations from intended interventions | Missing outcome data | Measurement of the outcome | Selection of the reported result | Overall Bias |
|---|---|---|---|---|---|---|
| Vera-Assaoka et al [32] | Low | Low | Low | Low | Low | Low |
| Sedano et al [11] | Some concerns | Low | Low | Low | Low | Some concerns |
| Sammoud et al [12] | Some concerns | Low | Low | Low | Low | Some concerns |
| Ramirez-Campillo et al [33] | Low | Low | Low | Low | Low | Low |
| Ramirez-Campillo et al [34] | Low | Low | Low | Low | Low | Low |
| Ramirez-Campillo et al [35] | Some concerns | Low | Low | Low | Low | Some concerns |
| Ramirez-Campillo et al [36] | Low | Low | High | Low | Low | High |
| Padrón-Cabo et al [37] | Low | Low | Low | Low | Low | Low |
| Nurper et al [38] | Some concerns | Low | Low | Low | Low | Some concerns |
| Negra et al [39] | Some concerns | Low | Low | Low | Low | Some concerns |
| Negra et al [14] | Some concerns | Low | Low | Low | Low | Some concerns |
| Michailidis et al [40] | Some concerns | Low | Low | Low | Low | Low |
| Jlid et al [41] | Some concerns | Low | Low | Low | Low | Some concerns |
| Hammami et al [42] | Some concerns | Low | Low | Low | Low | Some concerns |
| Hammami et al [9] | Some concerns | Low | Low | Low | Low | Some concerns |
| Drouzas et al [43] | Some concerns | Low | Low | Low | Low | Some concerns |
| Chtara et al [44] | Some concerns | Low | Low | Low | Low | Some concerns |
| Asadi et al [7] | Some concerns | Low | Low | Low | Low | Some concerns |
| Sammoud et al [12] | Low | Low | Low | Low | Low | Low |
| Liu et al [45] | Low | Low | Low | Low | Low | Low |

This study incorporated data from 14 randomized controlled trials involving 415 participants and compared the effects of plyometric training with regular soccer training on the change of direction (COD) ability of adolescent soccer players. The analysis included three COD ability indicators: the Illinois agility test, zig-zag run, and T-test (Fig 6). The results indicated that plyometric training significantly improved COD ability compared with regular soccer training, with a moderate positive overall effect size (SMD = -0.76, 95%CI: [-1.04, -0.47], p < 0.001). Planned subgroup analyses further revealed that plyometric training had a significant positive impact on the Illinois agility test (SMD = -0.71, 95% CI: [-0.98, -0.43], p < 0.001). However, no significant improvements were observed for the zig-zag run (SMD = -0.38, 95% CI: [-1.05, 0.29], p = 0.267) or the T-test (SMD = -1.47, 95% CI: [-3.36, 0.42], p = 0.127) compared with regular soccer training.

## Discussion

This systematic review and meta-analysis aimed to directly compare the effectiveness of plyometric training with regular soccer training in enhancing the jumping, sprinting, and change of direction abilities of adolescent soccer players. Existing evidence suggests that plyometric training provides significant benefits for adolescent athletes by offering high-intensity and non-regular training stimuli. A well-designed and scientifically supervised plyometric training program, when combined with regular soccer training, has the potential to achieve greater improvements in athletic performance compared with soccer practice and matches alone.

### Jumping performance

The systematic review and meta-analysis results indicated that plyometric training significantly enhances the jumping, sprinting, and COD abilities in adolescent soccer player. These findings align with previous meta-analyses [6,46–48]. For instance, Oliver et al. [46] reported that plyometric training produces statistically significant small-to-moderate

**Fig 3. Visualization of Publication Bias, Egger's Test, and Trim-and-Fill Adjustment.**

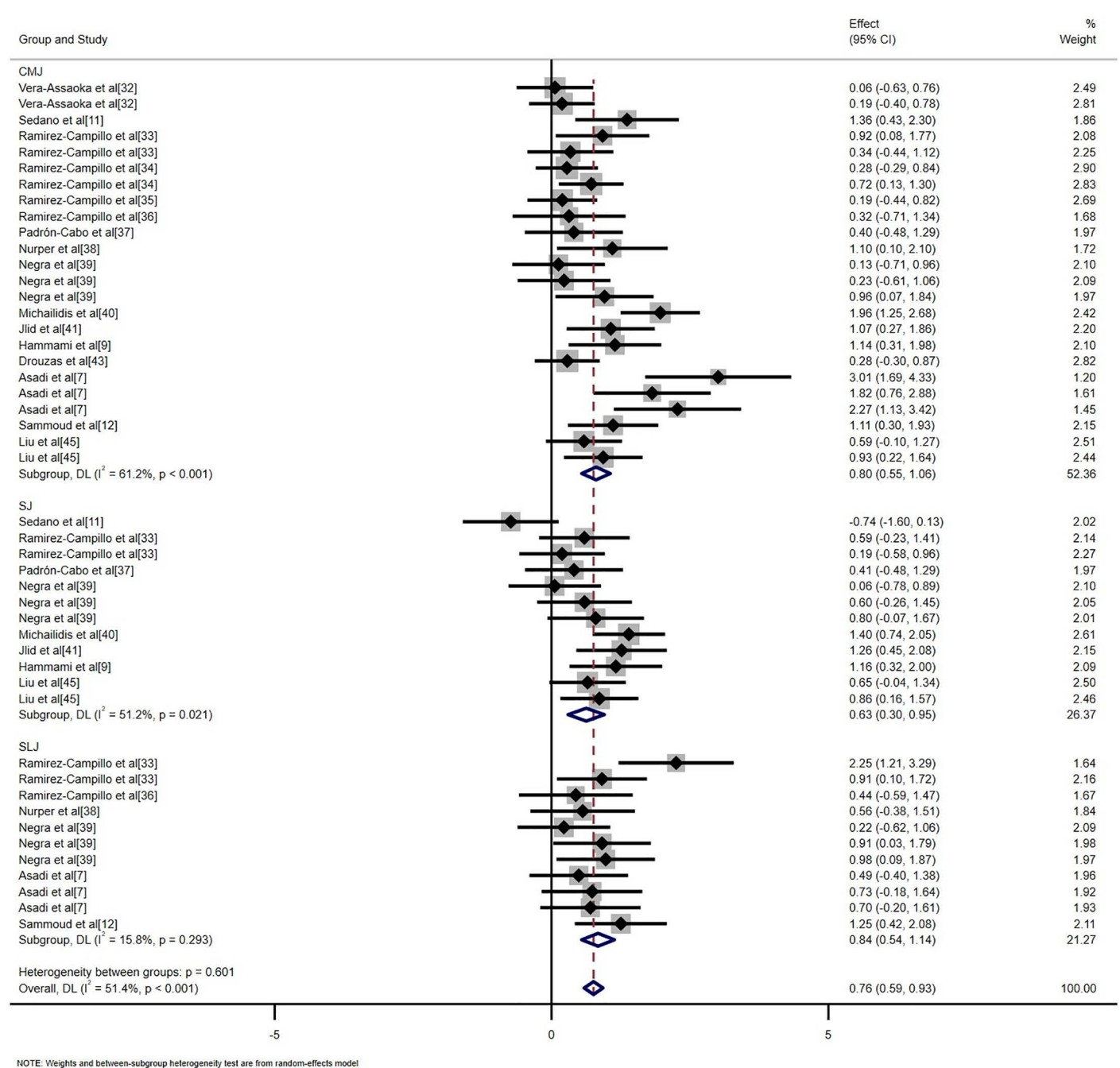

**Fig 4. Forest plot of the effects of plyometric training on jumping ability in adolescent soccer players.**

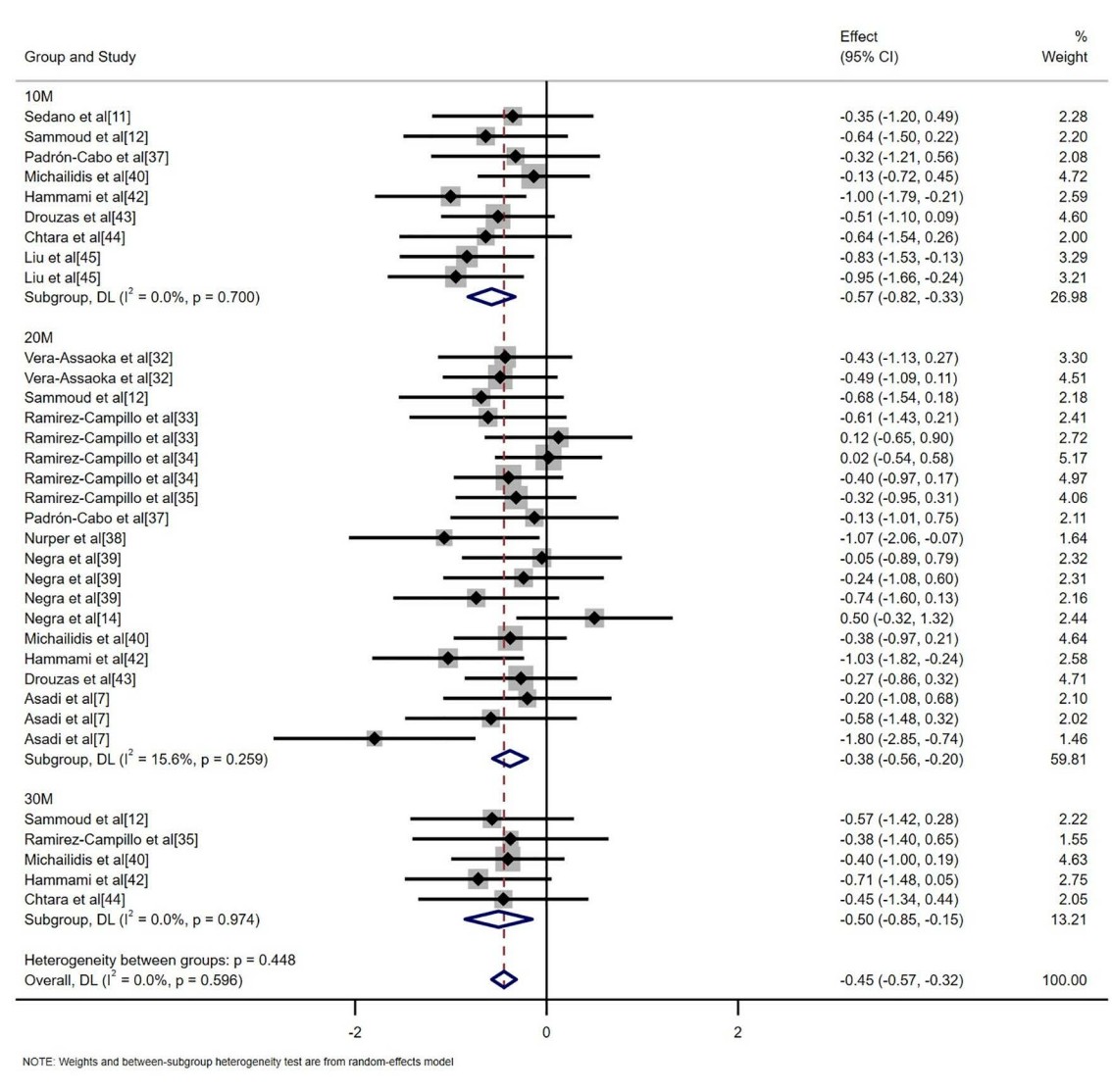

**Fig 5. Forest plot of the effects of plyometric training on sprinting ability in adolescent soccer players.**

improvements in jumping, sprinting, and COD abilities among highly trained adolescent soccer players (Hedges' g = 0.42–1.01). Recent studies have demonstrated significant correlations among jumping, acceleration, and COD abilities, suggesting that these capacities are interrelated and can be simultaneously improved through targeted training interventions [49,50]. This relationship is likely driven by shared neuromuscular adaptation mechanisms and biomechanical patterns, facilitating transfer effects across different athletic objectives [46]. This may suggest that plyometric training can simultaneously improve athletes' jumping, sprinting, and change-of-direction performances through similar neuromuscular adaptations [4]. Specifically, these neuromuscular functional adaptations include an increased neural drive to the agonist's muscles, alterations in muscle activation strategies, changes in the mechanical properties of the plantar flexor muscle-tendon complex, and modifications in muscle size and/or structure [51].

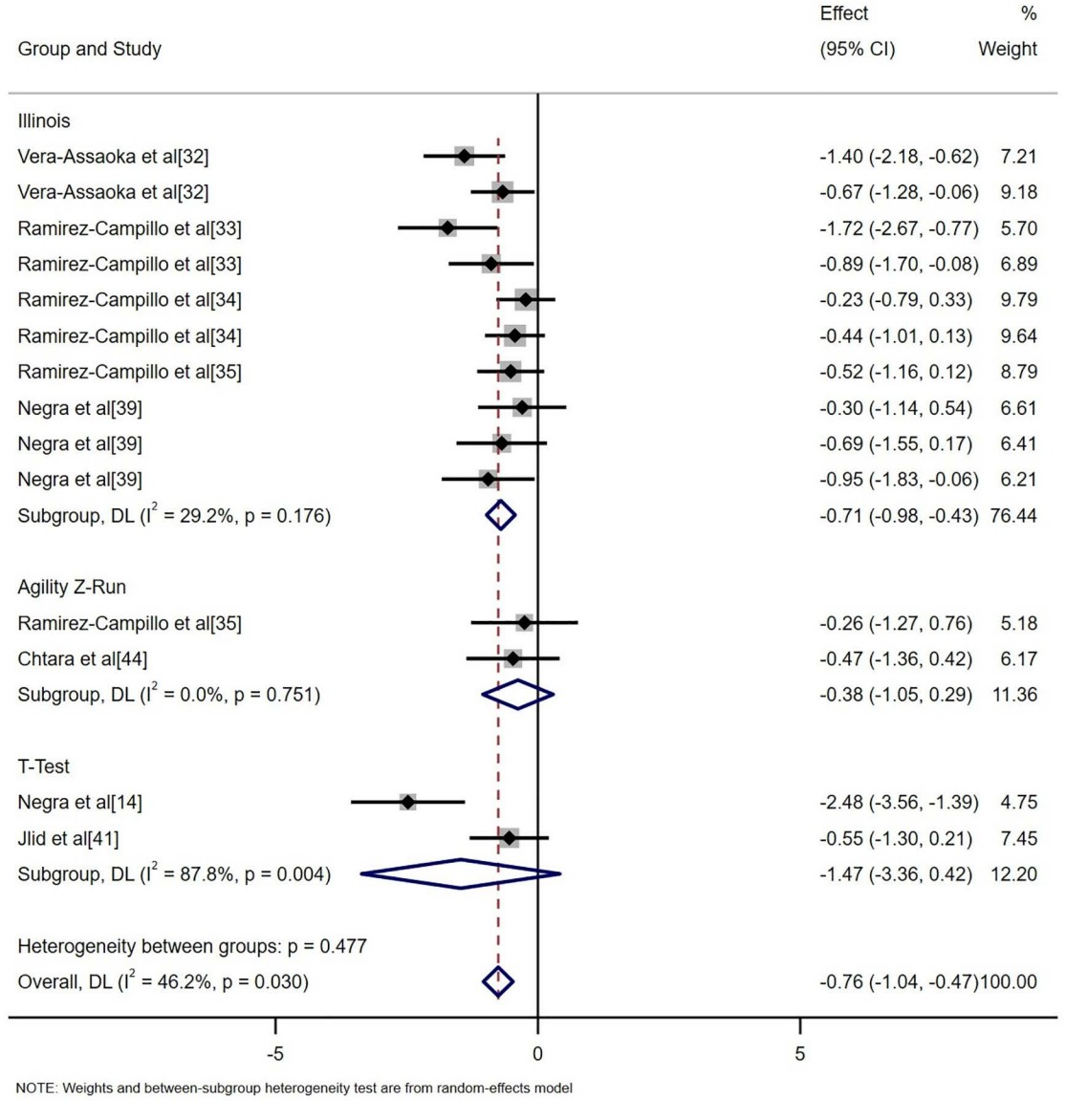

|  | Effect | % |
|---|---|---|
| Group and Study | (95% CI) | Weight |

**Illinois**

| Study | Effect (95% CI) | Weight |
|---|---|---|
| Vera-Assaoka et al[32] | -1.40 (-2.18, -0.62) | 7.21 |
| Vera-Assaoka et al[32] | -0.67 (-1.28, -0.06) | 9.18 |
| Ramirez-Campillo et al[33] | -1.72 (-2.67, -0.77) | 5.70 |
| Ramirez-Campillo et al[33] | -0.89 (-1.70, -0.08) | 6.89 |
| Ramirez-Campillo et al[34] | -0.23 (-0.79, 0.33) | 9.79 |
| Ramirez-Campillo et al[34] | -0.44 (-1.01, 0.13) | 9.64 |
| Ramirez-Campillo et al[35] | -0.52 (-1.16, 0.12) | 8.79 |
| Negra et al[39] | -0.30 (-1.14, 0.54) | 6.61 |
| Negra et al[39] | -0.69 (-1.55, 0.17) | 6.41 |
| Negra et al[39] | -0.95 (-1.83, -0.06) | 6.21 |
| Subgroup, DL ($I^2$ = 29.2%, p = 0.176) | -0.71 (-0.98, -0.43) | 76.44 |

**Agility Z-Run**

| Study | Effect (95% CI) | Weight |
|---|---|---|
| Ramirez-Campillo et al[35] | -0.26 (-1.27, 0.76) | 5.18 |
| Chtara et al[44] | -0.47 (-1.36, 0.42) | 6.17 |
| Subgroup, DL ($I^2$ = 0.0%, p = 0.751) | -0.38 (-1.05, 0.29) | 11.36 |

**T-Test**

| Study | Effect (95% CI) | Weight |
|---|---|---|
| Negra et al[14] | -2.48 (-3.56, -1.39) | 4.75 |
| Jlid et al[41] | -0.55 (-1.30, 0.21) | 7.45 |
| Subgroup, DL ($I^2$ = 87.8%, p = 0.004) | -1.47 (-3.36, 0.42) | 12.20 |

Heterogeneity between groups: p = 0.477
Overall, DL ($I^2$ = 46.2%, p = 0.030)   -0.76 (-1.04, -0.47) 100.00

NOTE: Weights and between-subgroup heterogeneity test are from random-effects model

**Fig 6. Forest plot of the effects of plyometric training on change of direction ability in adolescent soccer players.**

Furthermore, subgroup analysis revealed that plyometric training, when supplemented with regular soccer training, significantly improves the CMJ (ES = 0.8) and SLJ (ES = 0.84) performance in adolescent soccer players but only moderately enhances the SJ (ES = 0.63). This difference might occur because plyometric exercises primarily enhance the SSC, which does not fully target pure concentric contraction capabilities, as measured in the SJ test [10,52]. Consequently, plyometric training is more effective in jump tests involving SSC, while its enhancement in pure concentric jump tests is comparatively limited [10,52]. In contrast, resistance training, which involves less SSC, is more effective in improving SJ

performance [53]. Therefore, the choice of training modality should be carefully considered based on specific athletic performance requirements. Notably, relying solely on regular soccer training has failed to significantly enhance the jumping performance of adolescent soccer players, likely due to insufficient neuromuscular stimuli to elicit meaningful adaptations [11,12,40,43]. In contrast, studies have demonstrated that plyometric training, by providing additional neuromuscular stimuli, can significantly improve jumping performance within just 4–7 weeks [50,51]. These additional stimuli trigger a series of neuromuscular adaptations, including enhanced neural drive to active muscles, improved muscle activation strategies, altered mechanical properties of the plantar flexor muscle-tendon complex, and changes in muscle size and structure [51]. Therefore, incorporating structured plyometric training into regular soccer training is essential to specifically enhance explosive power and optimize the stretch-shortening cycle (SSC).

## Sprint performance

In comparison with regular soccer training, our findings indicated that while plyometric training significantly improves sprinting capabilities, the magnitude of enhancement is relatively modest (ES = 0.45), consistent with previous studies by Oliver et al. [46] and Chen et al. [54]. Oliver et al.'s meta-analysis [46] reported moderate improvements in acceleration (g = 0.74) and smaller effects on sprint speed (g = 0.42). Similarly, Chen et al. [54] observed reductions of 0.04 seconds in acceleration time and 0.12 seconds in sprint time, further corroborating the modest impact of plyometric training on sprint performance. This moderate effect may stem from neuromuscular adaptations, including improved joint proprioception and motion sensitivity through repetitive activation of articular mechanoreceptors during plyometric exercises [55]. The observed improvements in sprint performance may also be attributed to the plyometric training programs in the included studies, which universally integrated vertical and horizontal exercises [56]. Horizontal force predominates during the acceleration phase (≤10 meters), while vertical force becomes increasingly critical during the maximum velocity phase [57]. By integrating these two types of plyometric training, performance across different sprint phases can be optimized [58]. Interestingly, Silva et al. [59] found that plyometric training did not significantly enhance the linear sprinting performance of adolescent athletes participating in team sports. This may be attributable to the substantial incorporation of stretch-shortening cycle (SSC) activities in their regular training and competitions, which largely overlap with the stimuli provided by plyometric training, diminishing its marginal benefits for speed enhancement [40]. In contrast, several studies have shown that regular soccer training, which often includes frequent short-distance maximal sprints, can contribute significantly to speed development [14,40,60]. However, for athletes with lower baseline strength or those whose regular training lacks high-intensity SSC activities, plyometric training may still play an important role [10].

Additionally, we observed a more significant improvement trend in 10-meter linear sprinting (ES = 0.57) following plyometric training, which is consistent with the findings of Ramirez-Campillo et al. [61]. This finding is significant in soccer, where over 90% of sprints occur within 20 meters, with short sprints of approximately 10 meters being especially frequent during matches [62]. Plyometric training involves large joint movements like CMJ and SLJ, relying on slow SSC mechanisms similar to the acceleration phase of 10-meter sprints with longer ground contact times (≥250ms) [63]. In contrast, the maximum velocity phase requires rapid SSC mechanisms and shorter ground contact times, emphasizing vertical over horizontal force production [63,64]. Therefore, plyometric training may be more effective for enhancing acceleration and change-of-direction than maximum sprint speed. Interestingly, unlike our findings, Sáez de Villarreal et al. [47] found that linear sprinting beyond 10 meters improved more in adult athletes. This discrepancy may be due to differences in muscle-tendon unit (MTU) stiffness between adolescents and adults. Leg stiffness correlates significantly with maximum speed but not acceleration. [65]. Aging induces structural changes in tendons and muscles, such as collagen fiber thickening and alignment, enhancing tendon stiffness and facilitating faster elastic energy transfer and efficient SSC [66–68]. Conversely, adolescents have lower MTU stiffness, which may better support energy storage and release during the acceleration phase, enhancing slow SSC activities [69]. In summary, speed training should be more specific, and coaches

can systematically optimize athletes' performance at different speed phases by strategically combining horizontal and vertical jumps and scientifically configuring fast and slow SSC exercises, thereby comprehensively enhancing speed capabilities.

### COD performance

Previous studies have found that plyometric training effectively improves the COD ability of adolescent soccer players, which aligns with our meta-analysis results [10,13,32–34]. We found that plyometric training led to moderate improvements in COD ability for these athletes (ES = 0.76) [33,34,70] However, previous studies also showed that athletes undergoing only regular soccer training did not exhibit significant improvements and even experienced declines in COD performance [10,13,32]. These findings further support incorporating independent plyometric training during the season to enhance key athletic performance [10,13,32–34,44]. The improvement in COD abilities may be associated with the changes in strength development following plyometric training or an increase in lower limb eccentric strength [70,71]. This enhanced strength enables athletes to decelerate and change direction more rapidly and effectively [70].

The lack of significant improvements in the zigzag run and T-Test observed in the subgroup analysis may be partly explained by including only pre-adolescent athletes in the analyzed studies. Compared to their mid- or late-adolescent peers, pre-adolescents have an underdeveloped neuromuscular system, lower levels of anabolic hormones, and less advanced neuromuscular coordination [72]. These factors may limit their adaptive response to SSC training stimuli, resulting in relatively modest gains. During the post-peak height velocity (POST-PHV) stage, adolescents undergo significant hormonal and neuromuscular changes, including elevated testosterone levels, growth hormone, and insulin-like growth factor. These promote muscle hypertrophy and shifts in muscle fiber composition [73,74]. Collectively, these adaptations enable POST-PHV adolescents to utilize the elastic energy mechanisms of SSC activities more effectively, enhancing training outcomes [75,76]. Consequently, the lack of these physiological and neural adaptations in pre-adolescents may explain the absence of statistically significant differences in the analysis. The meta-analysis by Asadi et al. [71] further supports this notion, finding that twice-weekly, 7-week, 1,400-repetition moderate-intensity plyometric training is sufficient to cause a significant enhancement in COD abilities. However, the adaptive response to plyometric training depends on the maturity status of adolescents. Compared to mid- and late-adolescents (ES = 0.95–0.99), pre-adolescents (ES = 0.68) experience only a modest improvement in COD abilities. However, the study by Vera-Assaoka et al. [32] found that after 7 weeks of plyometric training (twice weekly, 60 jumps per session, including two sets of 10 repetitions from heights of 20, 40, and 60 cm), early adolescents showed a moderate improvement in COD speed (ES = 0.9), while late adolescents exhibited only a minor improvement (ES = 0.28). This finding contrasts with the general trend reported in the meta-analysis by Asadi et al. [71], which suggests that mid- and late-adolescents typically show more excellent adaptive responses to plyometric training. This discrepancy may imply that differences in biological maturity lead to varying adaptive responses to plyometric training intensities among adolescent athletes [77].

Additionally, differences in the direction-specific focus of plyometric training methods may further contribute to the high heterogeneity and non-significant results. Negra et al. [14] implemented vertical and horizontal combined plyometric training, effectively enhancing horizontal and vertical force production. This approach aligns well with the T-Test biomechanical demands, which require acceleration, deceleration, and rapid directional changes. In contrast, Jlid et al.'s [41] multidirectional plyometric training, while designed to reflect the movement patterns of competition, lacked specificity, particularly in optimizing horizontal force. These methodological differences underscore the importance of designing plyometric training programs that align with the biomechanical demands of COD tasks, accounting for the interplay of forces required for acceleration, deceleration, and rapid directional changes. Future studies should explore optimizing training protocols by incorporating directional-specific exercises and tailoring programs based on participant characteristics, such as biological maturity, to reduce heterogeneity and enhance training effectiveness.

## Limitations

The sample in this study encompasses a broad age range of adolescents, reflecting the overall improvements in jumping, sprinting, and change-of-direction abilities achieved through plyometric training. Coaches and strength and conditioning trainers may consider systematically incorporating plyometric training into routine soccer training programs to enhance these key performance metrics in adolescent athletes. However, adolescents' adaptive responses are significantly influenced by biological maturity, particularly in the development of neuromuscular coordination, tendon stiffness, and hormonal responses. The lack of systematic assessments of biological maturity in the included studies may limit the ability to fully capture variations in adaptive responses across different developmental stages, potentially affecting the interpretation of training effects and the generalizability of the conclusions. Future research should strengthen assessments of biological maturity using standardized methods such as Tanner staging or skeletal age evaluation to understand adolescents' training adaptations better [78].

Additionally, the adaptive responses of adolescent athletes may also be modulated by variables such as playing position [79], age groups [80], and tactical demands [81]. These variables not only directly impact physical demands but may also influence the effectiveness of training adaptations. Future research should further explore the application of plyometric training in the context of these variables and integrate comprehensive analyses of training load and fatigue monitoring to optimize training strategies, enhance outcomes, and mitigate potential health risks [82,83].

## Conclusion

This study demonstrates that plyometric training significantly enhances key athletic performance components in adolescent soccer players, including jumping ability, change-of-direction speed, and sprinting velocity compared to soccer-specific training alone. Based on these findings, it is recommended to incorporate plyometric training as a supplement to regular soccer training to optimize athletic performance and promote the long-term development of adolescent athletes.

## Supporting information

**S1 Data. All included studies.**
(XLSX)

**S2 Data. Row Data.**
(XLSX)

**S1 Checklist. PRISMA 2020 checklist.**
(DOCX)

## Author contributions

**Conceptualization:** Xuan Zhang, Yichao Xiao.

**Data curation:** Tianjing Zheng, Zhilong Huang.

**Formal analysis:** Xuan Zhang, Yichao Xiao.

**Funding acquisition:** Xicai Luo.

**Investigation:** Xiaowen Liang.

**Methodology:** Tianjing Zheng, Yichao Xiao.

**Project administration:** Xiaowen Liang.

**Resources:** Runzhou Kong, Zhilong Huang.

**Software:** Runzhou Kong, Xiaowen Liang.

**Supervision:** Runzhou Kong, Yichao Xiao.

**Validation:** Runzhou Kong, Zhilong Huang, Xuan Zhang.

**Visualization:** Runzhou Kong, Xuan Zhang, Yichao Xiao.

**Writing – original draft:** Tianjing Zheng, Runzhou Kong, Yichao Xiao.

**Writing – review & editing:** Tianjing Zheng, Xicai Luo, Yichao Xiao.

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
