## [Decision Letter · Decision Letter 0]

11 Oct 2024

PONE-D-24-38516Enhancement of Jump, Sprint, and Change of Direction Abilities: The Efficacy of Plyometric Training in Adolescent Soccer Training—A Systematic Review and Meta-AnalysisPLOS ONE

Dear Dr. Xiao,

Thank you for submitting your manuscript to PLOS ONE. After careful consideration, we feel that it has merit but does not fully meet PLOS ONE’s publication criteria as it currently stands. Therefore, we invite you to submit a revised version of the manuscript that addresses the points raised during the review process.

The article presents a valuable research question, but some methodological concerns have been raised, particularly in the discussion of the results. Please take into account the reviewers' comments.

We look forward to receiving your revised manuscript.

Kind regards,

Filipe Manuel Clemente, PhD

Academic Editor

PLOS ONE

5. As required by our policy on Data Availability, please ensure your manuscript or supplementary information includes the following:

Reviewers' comments:

Reviewer's Responses to Questions

**Comments to the Author**

1. Is the manuscript technically sound, and do the data support the conclusions?

Reviewer #1: Yes

Reviewer #2: Yes

2. Has the statistical analysis been performed appropriately and rigorously? 

Reviewer #1: Yes

Reviewer #2: Yes

3. Have the authors made all data underlying the findings in their manuscript fully available?

Reviewer #1: Yes

Reviewer #2: No

4. Is the manuscript presented in an intelligible fashion and written in standard English?

Reviewer #1: Yes

Reviewer #2: Yes

5. Review Comments to the Author

Reviewer #1: Dear Authors,

Thank you so much for the opportunity to read and revise such an interesting manuscript.

Althought I did not detect any fatal issue in your manuscript, there are some changes that need to be made.

Bellow you can find my revisions in a line-by-line approach.

ABSTRACT:

L11-12: "Soccer, as one of the most popular sports globally, demands a high level of physical fitness from its participants.". Consider specifying what "high level of physical fitness" is, in the context of soccer to provide clarity.

L12-14: "Particularly for adolescent soccer players, who are in a critical period of physical development, effective training methods are essential to enhance their sport performance." Clarify whether "sport performance" refers to specific skills or overall athleticism to keep the focus narrow.

L14-16: "Plyometric Training, known for improving lower limb explosiveness, has been proven effective for adult athletes, yet its impact on adolescent soccer players remains insufficiently and systematically assessed." Please change "has been proven effective for adult athletes" to "has been extensively studied in adult athletes" for accuracy.

L17-20: "This study aims to systematically evaluate the comprehensive effects of plyometric training on adolescent soccer players' jumping, sprinting, and change-of-direction abilities, with the intention of providing an empirical basis for coaches and athletes to optimize training programs and enhance competitive performance.". Please add "improving" or "assessing" before "the comprehensive effects" to clarify the study's intent.

L21-23: "Eligible randomized controlled trials were identified through searches of PubMed, Web of Science, Scopus, and SPORTDiscus databases. Study quality was assessed using the PEDro scale...". Mention if any grey literature or clinical trial registries were searched to address potential publication bias.

L28-30: "The results indicated that plyometric training significantly positively affected the jumping ability (SMD=0.76, 95%CI: 0.59, 0.93), sprinting ability (SMD=-0.45, 95%CI: -0.57, -0.32), and change-of-direction ability (SMD=-0.76, 95%CI: -1.04, -0.47) of adolescent soccer players.". If the direction of the SMD for sprinting and change-of-direction needs to be negative (indicating improvement), ensure this is correctly interpreted for readers unfamiliar with effect size metrics.

L31-32: "Plyometric training is an effective training method to enhance the sport performance of adolescent soccer players.". Define what "sport performance" measures are you referring to in this context to avoid ambiguity.

INTRODUCTION:

The introduction section appears somewhat short when considering the scope typically expected in such reviews, particularly for a topic as complex as sports training, specifically plyometric training in adolescent soccer players. Please consider my general suggestions for improvement of the Intorduction:

1. While there is an attempt to set the context by discussing the popularity of soccer and the importance of physical fitness, it might benefit from further elaboration. For instance, discussing the global soccer landscape, trends in youth sports development, or the physiological demands unique to soccer could provide a richer backdrop for understanding why this research is crucial.

2. The introduction should ideally present a review of the literature that outlines what is currently known about plyometric training in sports, especially its application in soccer. This could include how previous studies have approached training methodologies, the evolution of training practices, and any gaps or controversies in the existing research that your study aims to address.

3. There's a mention of specific studies and their findings, which is good, but the introduction could more explicitly state the research gap. Why is another study needed? How does this study uniquely contribute to the body of knowledge? This could be expanded to not only justify the study but also to highlight the innovation or specific contribution your research intends to make.

4. The objectives are somewhat buried within the text. A clear, standalone statement of what this systematic review aims to achieve would enhance clarity. This could be framed around questions like: What specific aspects of plyometric training are under scrutiny? How does this review intend to measure success or effectiveness? While the introduction needs expansion, it's also important to maintain clarity. Each paragraph should build logically towards your research question or hypothesis.

Specific comments to the Introduction section:

L39-40: "Soccer, one of the world's most beloved sports, draws countless enthusiastic participants into its fold[1].". Specify the scope or context, e.g., "youth soccer" if this study focuses on adolescents.

L40-42: "Concurrently, to cater to the needs of various demographics, a multitude of soccer leagues have emerged, offering a platform for soccer enthusiasts to showcase their skills[2]." If not directly relevant to your study's focus, consider trimming this for conciseness.

L42-44: "Despite the vast array of soccer players with differing levels of athletic prowess and age groups, they all share a common trait: the necessity for a solid foundation of physical fitness to exhibit commendable performance on the field[2].". Clarify what "solid foundation of physical fitness" means in this context.

L45-47: "Specifically, during a typical soccer match, players are often seen performing frequent jumps, sprints, and changes in direction, which are undoubtedly the most fundamental and common movements in soccer[3, 4].". Specify if "frequent" refers to frequency within a game or as a training need.

L47-49: "Moreover, a team's average jumping height and sprinting ability are significantly correlated with their level of play and performance[2, 5]." Cite or provide data on this correlation if it's a claim derived from referenced studies.

L49-50: "High-level athletes tend to possess superior jumping, running, and change-of-direction abilities compared to their lower-level counterparts.". Add "typically" or "are reported to" for scientific accuracy.

L50-52: "Additionally, high-level professional league players or international players often exhibit higher levels of jumping, sprinting, and change-of-direction abilities during their adolescence compared to their peers[6]." Reword "during their adolescence" could be rephrased to "as adolescents" for clarity.

L53-54: "Therefore, it is particularly essential to cultivate and develop these abilities during the adolescent phase for young soccer players[7]." Emphasize why adolescence is crucial for skill development, e.g., growth spurts, neuromuscular development.

L55-56: "Plyometric training, an effective method for enhancing lower limb explosiveness, is widely employed across various sports disciplines[8]." Define or briefly explain "lower limb explosiveness" for readers unfamiliar with the term.

L56-61: "This training strategy ingeniously capitalizes on the Stretch-Shortening Cycle (SSC) [9], during which the muscle undergoes eccentric, isometric, and concentric contractions, effectively storing elastic potential energy[9]. When the muscle rapidly transitions from a lengthened state to a shortened state, this stored energy is released, enabling the muscle to generate immense force instantaneously, thereby significantly enhancing the athlete's explosive power[10, 11]." Consider simplifying or splitting this explanation for better readability.

L63-67: "Chen et al.[8] demonstrated that plyometric training positively influenced countermovement jumps (CMJ) and 20-meter sprint performance in adolescent soccer players of different developmental stages; however, their systematic review and meta-analysis only addressed CMJ and 20-meter sprint performance." Specify if "different developmental stages" refers to age or pubertal status.

L68-70: "Given the efficacy of plyometric training for adolescent soccer players and the absence of comprehensive analyses in other systematic reviews and meta-analyses, this study aims to systematically evaluate the combined impact of plyometric training on adolescent soccer players' abilities in jumping, sprinting, and changing direction." Reword "absence of comprehensive analyses" to "limited comprehensive analyses" for accuracy.

METHODS:

L81: "Boolean operators (OR, AND) were utilized in conjunction with a series of keywords..." Mention any language restrictions or inclusion of grey literature searches.

Line 111: "Inclusion criteria: The study population consists exclusively of adolescent soccer players, with the age range defined as 10 to 18.99 years old." Specify if this range includes both genders or if there's a gender-specific analysis planned.

L113: "Outcome measures should include indicators of jumping, sprinting, or change-of-direction abilities (either in full or in part)." Define what constitutes "in part" for clarity.

L123: "Should the data remain inaccessible before the publication of the article, the literature in question would be excluded." Mention if there's a threshold for the number of attempts or a specific time frame for contacting authors.

L142-144: "Publication bias risk... using the trim-and-fill method." Briefly explain the trim-and-fill method for readers unfamiliar with it.

L151-152: "The SMD values can be interpreted..." Provide a reference or explanation for the SMD interpretation criteria used.

RESULTS:

L168: "A total of 20 studies were included in this research, encompassing 28 sets of randomized controlled trials.” If "sets" refer to different interventions or comparisons within studies, clarify this term for unambiguous understanding.

L170: "...all of whom were adolescent soccer players (aged 10 to 18.99 years)." Specify if these ages represent the mean or range across studies, or if this was a criterion for inclusion.

L171-172: "The intervention for the experimental group consisted of plyometric training..." Mention if there was variance in the plyometric training protocols across studies.

L177: "The average PEDro scale score of all included studies was 7.55..." Provide context on what this score means in terms of study quality or reliability.

L180-181: "Visual inspection could not quantify publication bias, hence Egger's test was employed..." Clarify if "visual inspection" refers to the funnel plot and ensure Egger's test method is briefly explained for clarity.

L184-189: "The trim-and-fill method was applied..." Specify how many studies were imputed for sprinting ability if applicable or clarify if it was not necessarily due to no detected bias.

L204: "This study incorporated data from 47 sets of randomized controlled trials..." "sets" refer to specific interventions or subsets within studies? clarify.

L205-206: "...encompassed three jumping ability indicators: CMJ, squat jump SJ, and SLJ." Use consistent capitalization, e.g., "CMJ, SJ, and SLJ" for uniformity.

L208: "The findings of the study demonstrated a positive effect..." Mention if this positive effect is considered small, moderate, or large based on your earlier SMD criteria.

L210-212: "Subgroup analysis revealed that plyometric training positively influenced..." Specify if these subgroup analyses were planned or post-hoc.

L218-219: "The Impact of Plyometric Training on the Sprinting Ability..." Maintain consistent formatting with the previous section headers.

L220: "This study incorporated data from 34 sets of randomized controlled trials..." If different from the previous number (47), explain why there might be a discrepancy.

L225: "The results of the study indicate that plyometric training has a positive effect..." Mention the effect size interpretation for sprinting.

L235-236: "The Impact of Plyometric Training on the Change of Direction Ability..." Ensure this section aligns with the previous sections in terms of structure and detail.

DISCUSSION:

L251: "This systematic review and meta-analysis aimed to evaluate..." Clarify that it's not just plyometric training but its effectiveness compared to conventional soccer training.

L256-257: "...adolescent athletes may experience a more substantial improvement..." Provide a brief rationale for why plyometric might outperform conventional training.

L261: "The systematic review and meta-analysis results indicate that plyometric training significantly enhances..." Ensure consistency in verb tense, e.g., "indicated."

L262: "These findings are consistent with previous meta-analytic outcomes..." Ensure these references are correctly cited and formatted.

L264: "However, recent studies have identified significant correlations among these attributes..." Briefly explain how these correlations could lead to transfer effects.

L270-273: "Specifically, these neuromuscular functional adaptations include..." Perhaps link these specific adaptations to how they contribute to performance improvements.

L278: "This disparity may stem from the fact that..." Rephrase for clarity, e.g., "This difference might occur because plyometric exercises mainly enhance..."

L283: "Notably, previous research has shown that control groups engaging only in regular soccer training..." Add a brief note on why regular soccer training might not suffice for jumping improvements.

L288: "In comparison with traditional soccer training, our findings indicate..." Connect this more explicitly to why plyometric training would have this effect.

L293: "Prior research has suggested that vertical plyometric training does not enhance..." Consider adding why this might be the case or what has changed in the current meta-analysis.

L304: "...diminishing the marginal benefits of plyometric training for speed enhancement." Perhaps explain why there might still be some benefit, even if marginal.

L319-320: "Additionally, we observed a greater improvement trend in 10-meter linear sprinting..." Clarify why this specific distance matters in soccer.

L335: "Therefore, the mechanical and neuromuscular demands of plyometric training may shift more towards..." Discuss what this means for training programs in soccer.

L353: "The meta-analysis results indicate that compared to conventional soccer training, plyometric training is a more effective method..." This might be repetitive; consider merging with previous discussion or highlighting new insights.

L356: "...which is consistent with previous research findings..." Mention specific findings or studies to add depth.

L364: "Compared to mid- or late-adolescents, pre-adolescents have not yet fully matured..." Expand on how this developmental stage impacts plyometric training efficacy.

L372: "However, the study by Vera-Assaoka et al[48] found..." Highlight the contrast between this study and general findings more clearly.

CONCLUSION:

L383: "This systematic review and meta-analysis provide a comprehensive assessment..." Change "provide" to "provides".

L384: "revealing its positive impact on enhancing jumping, sprinting, and change-of-direction abilities." Mention briefly how significant these improvements are, if possible, e.g., "marked positive impact..."

L385-386: "The findings indicate that training programs incorporating plyometric exercises..." Specify what aspects of performance are improved to remind the reader of the scope.

L387: "...further supporting the efficacy of plyometric training as an auxiliary method..." Consider removing "further supporting" if this point has been firmly established in the discussion.

L389-391: "Moreover, it is recommended that strength and conditioning coaches adhere to the principle of training specificity..." Make the recommendation more actionable, e.g., "Coaches should tailor plyometric programs to match the demands of soccer, focusing on..."

L391-392: "...strategically incorporating plyometric exercises of varying directions and types of SSC..." Briefly explain what SSC means here for those who might not know, e.g., "(stretch-shortening cycle)."

L392-393: "Combining them with traditional resistance training to comprehensively enhance..." Highlight why this combination is beneficial.

LIMITATIONS:

Please move this section to immediately before the conclusion section (i.e., to the end of the Discussion section).

L396: "...the results reflect the general trend to some extent..." Clarify what "general trend" refers to. Does it mean improvement in performance?

L397-398: "...the training adaptation to plyometric exercises in adolescents is largely influenced by biological maturity." Maybe add a line to explain why biological maturity is important, e.g., "as it impacts physical development and capacity for adaptation."

L398-399: "...lack systematic assessments of biological maturity..." Suggest what kind of assessments could have been used for better clarity.

L400: "...a deficiency that limits our accurate understanding..." Explain how this affects the interpretation of results, e.g., "thereby potentially skewing our interpretation of training effectiveness."

L401: "...may affect the generalizability of the conclusions." Discuss briefly how this might specifically limit generalizability, perhaps in terms of different maturity stages responding differently to training.

L401-402: "Future research should place greater emphasis on and enhance the assessment of biological maturity...". Offer examples of how future studies might assess biological maturity, e.g., Tanner staging, skeletal age assessment.

L402-403: "...to foster a more comprehensive understanding..." State what this comprehensive understanding could lead to, like "implementing training programs that match developmental stages.".

I wish you a very good work.

All the best.

Reviewer #2: General comments

The aim of this review was to systematically evaluate the comprehensive effects of plyometric training on adolescent soccer players' jumping, sprinting, and change-of direction abilities, with the intention of providing an empirical basis for coaches and athletes to optimize training programs and enhance competitive performance. This is a study on an interesting research topic, utilizing a systematic review design; however, the methodological procedures for constructing the systematic review, as well as the information extraction process, must be critically improved.

Specific Comments

Please, consider the section-by-section revisions:

• Introduction: The authors specify the Chen et al. study, where they report CMJ and 20-meter sprint performance, but plyometric training is something broader. There are methods such as DSI (https://doi.org/10.3389/fspor.2024.1282214), RSI (https://peerj.com/articles/15609/) that should be clarified. Also, field-based tests should be expanded much further to 20 metres (https://link.springer.com/book/10.1007/978-3-031-03895-2), not neglecting the methodological differences between agility and COD. The introduction should end with the objective (and possibly the study hypotheses), never with concluding remarks (lines 68-75). Please rephrase.

• Methods:

o Literature strategy: The literature strategy should be registered on a platform such as INPLASY or PROSPERO. Please register and add the code in this subsection. Also, the search strategy is much more than keywords and Boolean operators. This section should report how many authors conducted the search, which intra- and inter-reliability. Also, the author makes no mention of the PRISMA statement, although he reports the flowchart and uses the PICO approach. Please clarify this point (see: https://peerj.com/articles/14381/)

o Eligibility Criteria: The description of the PRISMA and PICOS strategy must be provided beforehand in the search strategy section. The inclusion and exclusion criteria are shown in Table 1.

o Data Extraction: Couldn't this incur a selection bias? For example, you used CMJ and SJ, why don't you evaluate the horizontal jump? Are the t-test, Zig Zag Drill and Illinois test the most commonly used tests to assess COD? Are the 10, 20 and 30 metre sprint tests suitable for all tests?

o Methodological Quality and Risk of Bias: The PEDro scale was well applied. I only recommend that you add an analysis of the risk of bias.

o Statistical Analysis: This section is well described, but you should clarify the legend in figure 2 (formula).

• Results: Some adjustments should be considered. In the Methodological Quality and Risk of Bias Assessment subsection, some statistical tests were not included in the methodology. The presentation of the Forest Plot is clear and precise (figures 3 to 5). However, there are some independent variables presented that are neither reported in the methodology nor discussed. For example: early, late, before, after, fixed, OPT, 4-12 weeks, pre to post-PHV.

• Discussion: Following on from the previous comment, the authors should include in the discussion the independent variables they considered for the meta-analysis. The contexts in which they are analysed have yet to be described, not least because there are specificities that must be taken into account in each sport. Also, the limitations section should report on future prospects and practical applications. I have some difficulty understanding objective practical applications for this meta-analysis with regard to Stregnth and Conditioning (please see: Branquinho, L., Ferraz, R., & Marques, M. C. (2021). 5-a-Side Game as a Tool for the Coach in Soccer Training. Strength & Conditioning Journal, 43(5), 96-108.

• References: An upgrade should be considered to expand the practical applications and analysis contexts of each study.

6. PLOS authors have the option to publish the peer review history of their article (what does this mean? ). If published, this will include your full peer review and any attached files.

**Do you want your identity to be public for this peer review?** For information about this choice, including consent withdrawal, please see our Privacy Policy .

Reviewer #1: **Yes: ** Rui Miguel Fernandes Pereira da Silva

Reviewer #2: **Yes: ** José Eduardo Teixeira

---

## [Author Response · Author response to Decision Letter 1]

6 Nov 2024

Dear Editor,

Greetings!

I would like to begin by expressing my sincere gratitude to you and the reviewers for the time and effort invested in the review of my manuscript titled "Enhancement of Jump, Sprint, and Change of Direction Abilities: The Efficacy of Plyometric Training in Adolescent Soccer Training—A Systematic Review and Meta-Analysis."

The valuable feedback provided by the reviewers has been instrumental in enhancing the content and quality of my paper.

I am pleased to inform you that I have diligently revised the manuscript based on the reviewers' suggestions. Enclosed in this email, you will find the revised manuscript along with a detailed response to the reviewers' comments.

I have endeavored to address all the suggestions made by the reviewers thoroughly. Should there be any issues requiring further discussion, I am readily available for communication with you and the reviewers.

Thank you once again for your support and guidance. I look forward to your further feedback.

Best regards,

Yichao Xiao

---

## [Decision Letter · Decision Letter 1]

8 Dec 2024

PONE-D-24-38516R1Enhancement of Jump, Sprint, and Change of Direction Abilities: The Efficacy of Plyometric Training in Adolescent Soccer Training—A Systematic Review and Meta-AnalysisPLOS ONE

Dear Dr. Xiao,

Thank you for submitting your manuscript to PLOS ONE. After careful consideration, we feel that it has merit but does not fully meet PLOS ONE’s publication criteria as it currently stands. Therefore, we invite you to submit a revised version of the manuscript that addresses the points raised during the review process.

Although improvements have been made, some concerns remain for the reviewers. Please address them in your revisions.

We look forward to receiving your revised manuscript.

Kind regards,

Filipe Manuel Clemente, PhD

Academic Editor

PLOS ONE

Reviewers' comments:

Reviewer's Responses to Questions

**Comments to the Author**

1. If the authors have adequately addressed your comments raised in a previous round of review and you feel that this manuscript is now acceptable for publication, you may indicate that here to bypass the “Comments to the Author” section, enter your conflict of interest statement in the “Confidential to Editor” section, and submit your "Accept" recommendation.

Reviewer #1: (No Response)

Reviewer #2: All comments have been addressed

2. Is the manuscript technically sound, and do the data support the conclusions?

Reviewer #1: Partly

Reviewer #2: Yes

3. Has the statistical analysis been performed appropriately and rigorously?

Reviewer #1: Yes

Reviewer #2: Yes

4. Have the authors made all data underlying the findings in their manuscript fully available?

Reviewer #1: Yes

Reviewer #2: Yes

5. Is the manuscript presented in an intelligible fashion and written in standard English?

Reviewer #1: No

Reviewer #2: Yes

6. Review Comments to the Author

Reviewer #1: Dear Authors,

I want to congratulate you for the immense work you have done.

I used the Word document of your manuscript with track changes ON to make my revisions.

I changed some parts of the manuscript and left some comments and suggestions throughout the entire manuscript. The Word document is attached.

All the best.

Reviewer #2: This study aims to systematically evaluate the potential for improving the comprehensive effects of plyometric training on adolescent soccer players' jumping, sprinting, and change-of-direction abilities, with the intention of providing an empirical basis for coaches and athletes to optimise training programs and enhance competitive performance. This is a manuscript with a very interesting research topic with high practical applicability and a very robust design. However, some sections need minor revisions.

Title: Replace ‘Adolescent’ with ‘Youth’, as it allows a more comprehensive view of the different stages of football player development.

Summary:

- Present the background more concisely, prioritising 2 sentences: an introductory sentence and a second sentence with the objectives of the study. Change-of-direction (COD) tends to be a commonly used abbreviation in this context (put it in the abstract as you did in the full text). Also, mention what PT means in the first sentence. Pliometric training?

- In the results, present a qualitative assessment of the magnitude of the effects on SMD.

- Introduction: In the last chapter, expand on other indices such as CPPF, DSI, IMF variables (please check: https://doi.org/10.3389/fspor.2024.1282214). The effect of PT on these variables should be deepened.

- Materials and Methods:

- The methodology is unimpeachable. I only suggest that you add the authors who selected and extracted the information from the articles, as well as inter-observer reliability. A third author should be considered to resolve discrepancies in the interpretation and extraction of information.

- Table 2: Why the three field tests were selected to assess Jumping Performance, Sprinting Performance and Change of Direction Performance. What were the criteria for selecting these tests? In addition, it should be made clear whether the included studies should integrate all three performance dimensions (jumping, sprinting or COD) or just one of the three.

Results:

In fact the meta analysis is robust, however some RCT studies should have been included: (e.g. https://doi.org/10.1016/j.gaitpost.2023.06.025 or https://link.springer.com/article/10.1186/s13102-022-00592-1). Please consider the previous point so that the meta-analysis actually covers all the relevant research in this area.

Discussion:

- The study considers the effects of PT to improve the comprehensive effects of plyometric training on the jumping, sprinting and change of direction skills of adolescent footballers, with the intention of providing an empirical basis for coaches and athletes to optimise training programmes and improve competitive performance. However, the authors do not clarify how it can influence taking into account field position (https://peerj.com/articles/15609/), age group (https://doi.org/10.3389/fpsyg.2024.1447968), contexts or tactical issues (https://doi.org/10.3389/fpsyg.2024.1447968).

Discussion:

- As for future recommendations on the control of biological maturation, it would also be interesting to analyse the relationship between plyometric training with jumping, sprinting and COD performance in the context of monitoring training load (please, see: https://doi.org/10.3389/fphys.2022.832202) and well-being (https://doi.org/10.1371/journal.pone.0225522).

7. PLOS authors have the option to publish the peer review history of their article (what does this mean? ). If published, this will include your full peer review and any attached files.

**Do you want your identity to be public for this peer review?** For information about this choice, including consent withdrawal, please see our Privacy Policy .

Reviewer #1: **Yes: ** Rui Miguel Fernandes Pereira da Silva

Reviewer #2: **Yes: ** José Eduardo Teixeira

---

## [Author Response · Author response to Decision Letter 2]

13 Jan 2025

Submit the reply to the reviewers in the form of an attachment to the submission system.

---

## [Decision Letter · Decision Letter 2]

5 Feb 2025

Effects of Plyometric Training on Jump, Sprint, and Change of Direction Performance in Adolescent Soccer Player: A Systematic Review with Meta-Analysis

PONE-D-24-38516R2

Dear Dr. Xiao,

We’re pleased to inform you that your manuscript has been judged scientifically suitable for publication and will be formally accepted for publication once it meets all outstanding technical requirements.

Kind regards,

Filipe Manuel Clemente, PhD

Academic Editor

PLOS ONE

Additional Editor Comments (optional):

Reviewers' comments:

Reviewer's Responses to Questions

**Comments to the Author**

1. If the authors have adequately addressed your comments raised in a previous round of review and you feel that this manuscript is now acceptable for publication, you may indicate that here to bypass the “Comments to the Author” section, enter your conflict of interest statement in the “Confidential to Editor” section, and submit your "Accept" recommendation.

Reviewer #1: All comments have been addressed

Reviewer #2: All comments have been addressed

2. Is the manuscript technically sound, and do the data support the conclusions?

Reviewer #1: Yes

Reviewer #2: Yes

3. Has the statistical analysis been performed appropriately and rigorously?

Reviewer #1: Yes

Reviewer #2: Yes

4. Have the authors made all data underlying the findings in their manuscript fully available?

Reviewer #1: Yes

Reviewer #2: Yes

5. Is the manuscript presented in an intelligible fashion and written in standard English?

Reviewer #1: Yes

Reviewer #2: Yes

6. Review Comments to the Author

Reviewer #1: Dear Authors,

Thank you so much for your efforts in answering all my comments and suggestions. Now, your manuscript has improved substantially.

All the best.

Reviewer #2: After two extensive and sustained rounds of peer review, the authors responded to all the reviewer's requests. I therefore recommend accepting the manuscript in its present form.

Congratulations.

7. PLOS authors have the option to publish the peer review history of their article (what does this mean? ). If published, this will include your full peer review and any attached files.

**Do you want your identity to be public for this peer review?** For information about this choice, including consent withdrawal, please see our Privacy Policy .

Reviewer #1: No

Reviewer #2: **Yes: ** José Eduardo Teixeira

---

## [Editor Report · Acceptance letter]

PONE-D-24-38516R2

PLOS ONE

Dear Dr. Xiao,

I'm pleased to inform you that your manuscript has been deemed suitable for publication in PLOS ONE. Congratulations! Your manuscript is now being handed over to our production team.

Kind regards,

on behalf of

Dr. Filipe Manuel Clemente

Academic Editor

PLOS ONE